# Polar Chromosomes—Challenges of a Risky Path

**DOI:** 10.3390/cells11091531

**Published:** 2022-05-03

**Authors:** Kruno Vukušić, Iva M. Tolić

**Affiliations:** Division of Molecular Biology, Ruđer Bošković Institute, 10000 Zagreb, Croatia; tolic@irb.hr

**Keywords:** mitosis, mitotic spindle, prometaphase, chromosome congression, polar chromosomes, chromosome segregation, aneuploidy, tumors, spindle assembly, motor proteins, CENP-E, dynein, polar ejection force

## Abstract

The process of chromosome congression and alignment is at the core of mitotic fidelity. In this review, we discuss distinct spatial routes that the chromosomes take to align during prometaphase, which are characterized by distinct biomolecular requirements. Peripheral polar chromosomes are an intriguing case as their alignment depends on the activity of kinetochore motors, polar ejection forces, and a transition from lateral to end-on attachments to microtubules, all of which can result in the delayed alignment of these chromosomes. Due to their undesirable position close to and often behind the spindle pole, these chromosomes may be particularly prone to the formation of erroneous kinetochore-microtubule interactions, such as merotelic attachments. To prevent such errors, the cell employs intricate mechanisms to preposition the spindle poles with respect to chromosomes, ensure the formation of end-on attachments in restricted spindle regions, repair faulty attachments by error correction mechanisms, and delay segregation by the spindle assembly checkpoint. Despite this protective machinery, there are several ways in which polar chromosomes can fail in alignment, mis-segregate, and lead to aneuploidy. In agreement with this, polar chromosomes are present in certain tumors and may even be involved in the process of tumorigenesis.

## 1. Introduction—Chromosome Congression and Alignment

The aim of the mitotic process is to segregate the genetic material packed into duplicated chromosomes equally between two daughter cells. To accomplish this, cells form a highly dynamic yet robust structure called the mitotic spindle [1,2,3]. In the majority of higher eukaryotes, chromosomes attach to spindle microtubules (MTs) by kinetochores, large macromolecular complexes that assemble specifically on the centromere of each chromosome [4]. The establishment of stable connections between kinetochores and MTs occurs during prometaphase, a period of mitosis defined by the nuclear envelope breakdown (NEBD) and full alignment of the chromosomes in the equatorial plane of the spindle, which defines the beginning of metaphase [5]. During early prometaphase, the cell shape changes, the interphase array of MTs is reorganized, and MT dynamics drastically increase as MTs invade an opened nuclear space packed with chromosomes [6]. This initiates interactions between MTs and kinetochores, resulting in the formation of the mitotic spindle and the alignment of chromosomes to the spindle equator [7].

The final positive outcome of prometaphase from the perspective of one chromosome is the stable attachment of sister kinetochores to MTs that emanate from opposite spindle poles and form kinetochore fibers (k-fibers), which is known as amphitelic attachment [8]. From the perspective of a cell, the final outcome is a stable attachment of all kinetochores to MTs and stable alignment of all chromosomes at the equatorial plane of the spindle that leads to the formation of the metaphase plate. Cells with normal metaphase plates are prepared to quickly initiate the movement of separated chromatids to their respective spindle poles during anaphase [9,10]. Kinetochore interaction with MTs is monitored by the spindle assembly checkpoint (SAC), complex signaling machinery that prevents the onset of anaphase if chromosomes with inadequate attachments are detected [11]. The main erroneous attachments include syntelic attachment, where both sister kinetochores interact with MTs emanating from the same spindle pole, and merotelic attachment, where a single kinetochore is connected to both spindle poles [12]. If improper kinetochore attachments are not resolved by error correction mechanisms [13], the outcome is often chromosome mis-segregation and aneuploidy, a state of chromosome number that is not a multiple of a haploid complement [14], both of which are associated with multiple congenital diseases and various types of cancers [15].

The aim of this review is to present classic and novel models of chromosome alignment together with various biophysical and molecular aspects of spindle biology involved in this process, with a special emphasis on polar chromosomes positioned close to the spindle poles. We discuss how passage close to the pole represents a regular alignment route of certain chromosomes during prometaphase and discuss why polar chromosomes are observed even during later stages of mitosis in cells characterized by various alignment defects. Furthermore, polar chromosomes are characterized by alignment that is dependent on the activity of kinetochore motors and a transition from lateral to end-on attachment of kinetochores to MTs, all of which can result in delayed alignment and an increased chance of acquiring merotelic attachments. As prometaphase is highly variable between organisms, we focus on data from vertebrates, in particular human cells, but also describe key concepts that were frequently developed in a variety of different model organisms.

## 2. Different Routes to Chromosome Biorientation—Curios Case of Polar Chromosomes

Even though the aim of chromosomes during prometaphase is the same, namely biorientation and alignment, they can achieve this aim by different spatial paths that are mainly defined by the initial position of the chromosome with respect to the spindle during early prometaphase. This phenomenon is related to the fact that chromosome positioning with respect to main nuclear axes is generally transmitted from interphase to telophase in human cells [16]. As different spatial chromosome pathways at the beginning of mitosis are characterized by distinctive requirements for kinetochore motors [7,17], we consider two regions in which chromosomes can be positioned at NEBD, central and peripheral polar (Figure 1A). Central chromosomes positioned close to the nascent spindle are efficiently captured by MTs from both centrosomes, thus achieving rapid biorientation [17,18] (Figure 1A, route 1), independently of kinetochore motors [19]. Central chromosomes that are not located within the nascent spindle are transported directly to the spindle equator, where they become rapidly bioriented (Figure 1A, route 2) or first undergo a poleward-directed movement on the emerging spindle before initiating congression towards the equatorial plane [20,21,22] (Figure 1A, route 3). These movements depend on the kinetochore motors to a different extent [23]. Lastly, peripheral polar chromosomes, comprising 10–20% of all chromosomes, first move poleward before initiating congression towards the equatorial plane (Figure 1A, route 4), which strongly depends on kinetochore motors [7,18].

There is confusion in the literature regarding the usage of the terms polar, peripheral, mono-oriented, unaligned, and misaligned chromosomes. We suggest distinct terms for prometaphase and metaphase chromosomes because certain configurations are normal transient stages during prometaphase, whereas in metaphase, they represent errors. In prometaphase, it is important to discriminate different spatial origins of chromosomes, and in metaphase, the attachment status of chromosomes needs to be taken into account. Thus, a suitable term for peripheral chromosomes located closer to one pole during prometaphase, which are dependent on kinetochore motors, is ‘peripheral polar chromosomes’ [18,24]. In certain cases, at the end of spindle elongation, not all chromosomes are aligned in the equatorial plane, which is commonly called pseudo-metaphase [25] (Figure 1B). For chromosomes located close to the spindle poles during pseudo-metaphase, which are rarely bioriented and can achieve various types of attachment to MTs [25], we argue that the most suitable term is ‘unaligned chromosomes’ [26] (Figure 1B). For chromosomes placed outside of the metaphase plate but still closer to it than to the pole, which are usually bioriented and show oscillatory behavior [27], the term ‘misaligned chromosomes’ is appropriate (Figure 1B). The term ‘moonoriented chromosome’ that was commonly used in older mitosis literature denotes the chromosome that is closer to one pole, either facing the equatorial plane or the spindle periphery, during prometaphase or pseudo-metaphase [7,28]. We refer to chromosomes that come close to the spindle pole during prometaphase and pseudo-metaphase as polar chromosomes, and they are the focus of this review.

## 3. General Models of Chromosome Alignment

### 3.1. Search-and-Capture Model

At the heart of the alignment of every chromosome and the spindle assembly, in general, is the classic “Search-and-Capture model” (S&C) [29]. According to this model, which represents the first conceived and experimentally demonstrated mechanism of chromosome alignment [30], astral MTs nucleated from the centrosomes invade the nuclear space and stochastically interact with chromosomes. The fast transition between the growing and shrinking states of centrosomal MTs at their plus ends, known as dynamic instability, enables MTs to ‘search’ the cytoplasmic space in various directions until they ‘capture’ and establish a stable connection with kinetochores [31] (Figure 2A, part 1). Although S&C model stood the test of time and remained the most relevant model of chromosome alignment to date, it was substantially modified in the last decades [17]. First, mathematical modeling of S&C with real-life parameters revealed that the duration of mitosis would exceed the experimentally observed times by several orders of magnitude [32,33]. Therefore, additional mechanisms that accelerate MT search and kinetochore capture have been proposed, including regulated changes in cell shape occurring during early prometaphase [34], prepositioning of the main components of the spindle [21], guided growth of astral MTs towards the kinetochore [35], nucleation of kinetochore MTs [20], pivoting of astral MTs around the spindle pole [36], rotation of chromosomes [32,37,38], and kinetochore expansion before MT capture [22] (Figure 2A, parts 2–5) (see detailed review by [17]).

### 3.2. Oscillating at the Equator—Maintenance of Alignment

Kinetochores that become stably aligned maintain their position within the metaphase plate by moving in an oscillatory manner around the equatorial plane [39,40]. The main mechanisms that maintain chromosome alignment in vertebrate cells include the regulation of k-fiber plus end dynamics by motors such as Kif18a/kinesin-8, sliding of bridging MTs, and the action of polar ejection forces (PEFs) [41,42,43,44,45,46], though the level of contribution of each mechanism is still unclear. The relationship between the mechanisms that drive chromosome congression to the mechanisms that maintain their alignment at the equator is also unclear [7]. However, since current alignment models rely on the presence of k-fibers at both sister kinetochores, while congression requires neither k-fibers nor a sister kinetochore [47,48,49,50], the concept that these mechanisms are mechanistically distinct is prevalent in the field. Regarding the function of kinetochore motors that influence chromosome congression movement, perturbation of CENP-E and chromokinesins can influence the maintenance of alignment of chromosomes at the equator [19,51,52,53], but it is not clear if this is related to the same mechanical causes as during congression.

## 4. Biomechanical and Molecular Aspects of Polar Chromosome Congression

### 4.1. Getting to the Spindle—Movements towards and across the Polar Region

The first movement typical for a polar chromosome is transport to the polar region of the forming spindle [54,55,56,57,58,59,60]. Two mechanisms have been proposed to drive this transport, one based on MTs and kinetochore motors and the other on actin (Figure 2). In the first mechanism, peripheral chromosomes are transported poleward by kinetochore-bound dynein walking along astral MTs and pulling kinetochores towards the MT minus end, which is close to the spindle pole [61] (Figure 2B). By studying the congression of chromosomes after depletion of dynein [18] or its kinetochore-specific adaptors ZW10 [61] and Spindly [62,63], it was found that the congression of about 20% of chromosomes is drastically impaired when dynein is absent from kinetochores. The maximum velocity of initial chromosome movements is comparable to the rate at which cytoplasmic dynein moves vesicles along MTs during interphase and is in the range between 10–20 μm/min in human cells [61,64]. For the majority of kinetochores in human cells, this fast movement is brief, and the overall displacement is about 1 μm [21]. Alternatively, it has been suggested that in the absence of dynein, depolymerization of astral MTs moves kinetochores with end-on tethered CENP-E motor poleward, though at a reduced velocity compared to dynein-driven movement [65] (Figure 2B).

Interestingly, rapid movements of chromosomes are often not directed toward the spindle pole as expected but rather to a position near the center of the nascent spindle [21] (Figure 1, route 2). Why some peripheral chromosomes move rapidly poleward while others move towards the future equatorial plane of the forming spindle remains incompletely understood. As most astral MTs have a minus-end in the centrosomes, the main MT nucleation sites of the cell [66,67], this would imply that chromosomes that come into contact with astral MTs should show characteristic poleward movement. This is probably the case for most polar chromosomes located behind the spindle pole [18], where astral MTs display unidirectional polarity with plus ends protruding toward the cell cortex and minus ends embedded in the centrosome [2]. However, in the central region away from the spindle, chromosomes often show surprisingly fast orthogonal movements of kinetochores by which they come close to the spindle equator [23], whose mechanism is not completely known.

As the majority of chromosomes approach the spindle region soon after NEBD, even in conditions that diminish the activity of kinetochore dynein [18,61], alternative models arose that tried to explain the characteristic synchronous spindle-directed movements of chromosomes during early prometaphase. Contractility of actin filaments on the remnants of the nuclear envelope during and after NEBD has been shown to reduce chromosome scattering during the early stages of spindle assembly [68,69] (Figure 2C). Defects in this contractility were associated with an increase in the proportion of unaligned chromosomes during pseudo-metaphase and an increase in chromosome mis-segregations during anaphase [68]. Although the contribution of actin contractility to initial chromosome movements in somatic cells is not completely clear, this mechanism is especially relevant to gathering the scattered chromosomes to a small space of the future spindle in larger cells, such as mammalian oocytes [70]. Large physical barriers, similar to the actin-based one that operates in oocytes, have been observed in somatic cells, including the perinuclear actin cage in epithelial cells [71], and less rigid barriers often termed the ‘spindle matrix’ derived from the nuclear envelope in other cell types [72,73], which can contribute to the synchronous gathering of peripheral chromosomes on the spindle.

An interesting aspect of the characteristic movements of peripheral polar chromosomes during prometaphase is their passage across the large polar region [21,74,75]. Contrary to the poleward movement of chromosomes after NEBD, the movement across the pole cannot be explained by minus or plus end directed movement of the kinetochore along a single astral MT, either end-on or laterally attached, and would thus require mechanically different mechanisms. This movement should be directed towards the metaphase plate, as this is the general direction of congression [7], and it would be interesting to explore what defines the directionality of these movements and how they depend on different types of kinetochore-MT attachments.

### 4.2. Getting to the Equator—Congression from the Spindle Pole to the Spindle Midplane

The most studied movement of polar chromosomes during prometaphase is the process of transport of chromosomes from the spindle pole to the spindle equator, i.e., congression. The congression follows some basic principles defined by the knowledge gained through decades of studying this process. First, electron microscopy (EM) and confocal microscopy images of early prometaphase spindles revealed that kinetochores that are located between the spindle pole and the equator are in direct lateral contact with the walls of MTs, whereas end-on attachments are rarely observed [20,21,22,47] (Figure 3A, part 1). It thus became widely accepted that lateral attachments often precede biorientation and formation of amphitelic attachments [47,50,65]. Second, one of the most prominent features of congression is the gradual increase in interkinetochore distance, with and without k-fibers [21] (Figure 3A, parts 1 and 2). Third, kinetochores change their orientation with respect to the axis of the forming spindle from random to parallel orientation, independently of the formation of end-on attachments [21] and without large displacement and interkinetochore stretching [22] (Figure 3A, parts 1 and 2). Lastly, chromosomes are able to congress efficiently after ablation of chromosome arms or of one kinetochore [35,76], suggesting that neither forces acting on chromosome arms nor tug-of-war between sister kinetochores are prerequisites for chromosome congression (Figure 3A, parts 3 and 4).

A large amount of work revealed that The essential factor responsible for chromosome congression is the CENP-E [18,19,25,52], a plus end directed kinesin from the kinesin-7 family [77]. CENP-E is localized to the fibrous corona, a transient structure present adjacent to the outer kinetochore layer in the absence of kinetochore end-on attachments to MTs during prometaphase [78,79,80] (Figure 3), similar to dynein [81]. Currently, the dominant model for congression involves CENP-E-driven kinetochore gliding laterally alongside preformed MTs [47,82] (Figure 3B). Due to the opposite walking directionalities of CENP-E and dynein, the situation when they localize on the same kinetochore could result in a tug-of-war that would prevent chromosome alignment because dynein would pull the kinetochore poleward, while CENP-E would pull the same kinetochore towards the equatorial region. However, this does not occur because of tubulin detyrosination, a post-translational modification enriched on long-lived MTs such as k-fibers, that up-regulates CENP-E activity, making it a dominant force acting on the kinetochore in this region [83,84] (Figure 3B).

Based on the gliding activity of CENP-E, it is assumed that this motor delivers peripheral polar chromosomes to the area of the spindle that is favorable for the formation of proper end-on attachments [17]. However, it is still not clear what would define the region where biorientation is favorable. A recent model proposed that this region is defined by the extent of antiparallel overlaps of interpolar MTs [23]. In this model, transient interactions between short MTs protruding from the kinetochore, and antiparallel MTs of the spindle, help to organize kinetochore MTs into two bundles that are oriented with the minus ends facing toward the opposite spindle poles (Figure 3C). An important feature of this model is that CENP-E sorts the plus ends of MTs and gathers them at the kinetochore [23] (Figure 3C). However, it is currently unclear how CENP-E function in gliding along MTs is coordinated with its function in sorting MT ends and whether one excludes the other. Because of the synchronous bi-orientation of kinetochores in human cells during prometaphase [23], the model of kinetochore-nucleated MTs also proposes that the process of chromosome biorientation is more deterministic than is usually assumed [17].

This model also implies the role of the Ran-GTP gradient centered on chromatin (Figure 2A) and the chromosomal passenger complex (CPC) on the kinetochore [85,86], which help the nucleation of MTs from kinetochores. Still, while it was shown that the Ran-GTP gradient could attract astral MT growth [35], it is not clear to what extent the Ran-GTP gradient affects the growth of short MTs from kinetochores. The interaction of these MTs with antiparallel bundles, although an important part of the model, is not essential since biorientation is only moderately delayed after depletion of the main crosslinker of antiparallel MTs PRC1 [23]. Strong connections between antiparallel bundles and k-fibers of sister kinetochores are observed during metaphase in vertebrate cells when almost every pair of sister kinetochores is linked with a bundle of antiparallel MTs called the bridging fiber [44,87,88]. However, the exact coordination of antiparallel bundle formation and chromosome congression is not fully understood, but interestingly, both are regulated by CENP-E [18,89]. Furthermore, the mechanism by which small MTs protruding from the kinetochore would quickly grow into a fully mature k-fiber, and the role of dynein in this process is unclear (Figure 3C), although MT growth from the kinetochore was observed after laser ablation in Drosophila S2 cells [90,91]. Similarly, in human cells, the severed k-fibers quickly reincorporate into the spindle by a dynein-mediated mechanism that connects the disconnected k-fibers with neighboring MTs [92]. Yet, without dynein, the formation of k-fibers is not disrupted [92]. Amplification of short and sorted kinetochore-nucleated MTs could be aided by the augmin complex [93] that contributes to kinetochore MT growth even in the absence of pre-existing centrosomal MTs [94].

An alternative model of CENP-E activity postulates that this motor is required for the tethering of kinetochores to the lateral sides of MTs [95] as a first step in the lateral to end-on conversion [96] (Figure 3D). CENP-E is enriched on laterally bound kinetochores [97]. During this gradual process, laterally attached kinetochores rarely detach in the presence of CENP-E, while kinesin-13 MCAK is required for efficient removal of kinetochore attachments to the lateral walls of the MT (Figure 3D) [96]. Since the vertebrate kinetochore is bound by 10–30 MTs [98,99,100], both the wall-tethering and the gliding model of the CENP-E function agree that end-on conversion is likely to be a gradual process in which some MTs at first remain bound laterally to kinetochores, while some become end-on attached, known as mixed lateral end-on attachment [96] (Figure 3B). A list of potential functions of kinetochore-associated CENP-E does not end here, as CENP-E has also been demonstrated to be a processive bi-directional tracker of dynamic MT tips [53], CENP-E can regulate the SAC through its interaction with BubR1 kinase [101], and both are important for stabilization of kinetochore–MT end-on attachments. In conclusion, while CENP-E function is indispensable for the congression of polar chromosomes, it is still not clear what the dominant mechanisms of CENP-E function are in human cells.

### 4.3. How Do Polar Chromosomes Set Their Distance to the Spindle Pole?

In addition to the role of dynein in the initial poleward motion of kinetochores [61], dynein depletion was reported to increase the number of polar chromosomes with stabilized end-on MT attachments and mature k-fibers in bipolar spindles [18]. Thus, the poleward movement of peripheral chromosomes mediated by dynein is not only a fundamental part of the chromosome pathway to the equator, but it represents a mode of preventing premature stable kinetochore-MT end-on attachments in the polar region of the spindle [102] (Figure 4A), as polar chromosomes would otherwise be particularly prone to erroneous syntelic attachments to the neighboring pole [54,103]. The molecular mechanisms that are responsible for the prevention of the premature establishment of end-on attachments, and implications of those for the alignment of polar chromosomes, are still under active investigation. To date, a role in destabilizing end-on attachments in the polar region has been proposed for Aurora A kinase in human somatic cells and oocytes [54,104] (Figure 4), with evidence for the role of Aurora B kinase in somatic cells [18,103].

As kinetochore proximity to the pole could be a critical factor for the stability of attachments, it is important to understand the mechanisms that determine the distance of chromosomes to spindle poles. Chromosome-associated kinesins, termed chromokinesins, including Kid and Kif4a, have been shown to be important in adjusting the distance at which chromosomes approach the spindle pole [18] (Figure 4A). Kid and Kif4a are thought to act through MT-based force that ‘eject’ chromosomes from poles, termed PEF, which works either through the pushing of chromosome arms by MT-polymerization [105] or by the activity of plus end directed chromokinesins, mainly Kid, that walks along the MT and tows chromosome arms [51,106] (Figure 4A). Thus, PEF opposes dynein activity close to the pole (Figure 4A). PEF is predicted to increase near the pole where the density of MTs is high and to scale with the size of the chromosome [43]. PEF is involved in the stabilization of MT attachments in this region [18,42,102,107], which could be because it pushes chromosomes outside of the Aurora A activity gradient [54] (Figure 4A) or because it increases the tension on the kinetochore. The latter possibility is consistent with the notion that polar chromosomes require constant tension away from the pole to establish stable kinetochore-MT attachments [108].

However, outside the polar region, the effect of PEF is probably minor in unperturbed mitosis and increases with reduction in kinetochore-based forces, as seen in monopolar spindles [109], when k-fiber formation is perturbed [110], or when major kinetochore motors are depleted [18]. Furthermore, polar chromosome fragments in human cells after laser ablation move in random directions [18], implying that PEF is not critical for congression exclusively toward the spindle equator. Similar conclusions were drawn from experiments including laser irradiation of a kinetochore [49] and mitosis with un-replicated genomes (MUGs) [35], in which kinetochores are required and sufficient for chromosome congression. Collectively, these experiments imply that the forces generated by kinetochore motors and kinetochore-MT attachments are dominant over PEFs for chromosome congression in unperturbed spindles, but the role of PEF could be very important close to the polar region.

The tug-of-war between different mechanisms that set the distance to the pole was nicely recapitulated in experiments in which the balance between different factors was changed in monopolar spindles [18,110] (Figure 4B). Monopolar spindles are often used as a model to study the behavior of polar chromosomes, but one should be cautious with interpretations because monopolar spindles are characterized by end-on attachments such as syntelic and monotelic [111], which are rare in unperturbed bipolar prometaphase spindles [21]. For example, in otherwise unperturbed monopolar spindles, chromosomes localize closer to the pole if CENP-E or chromokinesin activity is perturbed and farther away if dynein is depleted [18] (Figure 4B). CENP-E in this condition probably mediates the motion of the leading kinetochore [7]. On the other hand, if NDC80 is perturbed in monopolar spindles, lateral attachments become dominant [112,113], and in this case, perturbation of CENP-E activity increases the distances of kinetochore to poles, while it decreases after additional perturbation of chromokinesin Kid, similarly to bipolar spindles without k-fibers [110] (Figure 4B). These results imply that CENP-E plus end directed motor activity is dominant in conditions where stable MTs are present, such as during late prometaphase, while during early prometaphase, CENP-E could suppress chromosome congression by causing kinetochores to track short and unstable MTs [53,110] (Figure 4B). Stable MTs are also more detyrosinated, which favors CENP-E activity over dynein [83] (Figure 4A). Taken together, the distance of chromosomes to the spindle poles is an important factor that determines the stability of end-on attachments, probably due to the action of Aurora kinases, while the actual distance is determined by the tug-of-war between dominant kinetochore motors and the supportive action of PEFs close to the poles.

### 4.4. Significance of Centrosome Prepositioning during Prophase

The position of centrosomes prior to NEBD has been recognized as an important factor that influences mitotic fidelity [17]. Centrosome separation occurs in coordination with NEBD [75], by Eg5/kinesin-5 driven sliding of antiparallel MTs [111,114], with the help of additional players, including myosin II [69], actin [115], and nuclear and cortical dynein [116,117]. In non-transformed cells, centrosomes fully separate before NEBD, which is termed the prophase pathway (Figure 5A), whereas, in tumor cell lines, centrosomes often separate after NEBD, also known as the prometaphase pathway [21,74,118] (Figure 5B). Additionally, the internal signals provided by the nucleus and the cytoskeleton predominantly position the separated centrosomes on the shortest nuclear axis before NEBD [21,75,119]. Several papers revealed that the degree of centrosome separation upon NEBD plays an important role in determining the types of kinetochore-MT attachments that form in early prometaphase, with a larger number of erroneous attachments and mis-segregations in the case of delayed separation of centrosomes at NEBD, and the formation of a prometaphase rosette [74,120,121] (Figure 5B). It was proposed that delayed centrosome separation promotes syntelic attachment in the vicinity of one pole, which can develop into merotelic attachment by an additional MT approach from the other pole [120] (Figure 5B, part 1). Similar results have been reported after bipolarization of a monopolar spindle [56]. Furthermore, the geometry of the spindle during metaphase is often asymmetric after delayed or accelerated centrosome separation, which could also lead to the formation of merotelic attachments due to the altered angle of the MT approach from a more distal centrosome [121,122,123] (Figure 5B, part 2). Merotelic attachments are particularly error-prone, as they can bypass error correction and SAC mechanisms [12,55,56]. A broadly similar role of centrosome separation has been reported in coalescing multipolar spindles [124].

Additionally, pre-NEBD centrosome separation places most chromosomes between the two separated spindle poles, where alignment to the metaphase plate is rapid [5,17]. Thus, incompletely separated centrosomes increase the proportion of peripheral polar chromosomes at NEBD, consequently placing more chromosomes in non-preferential positions where alignment and biorientation are slower. Therefore, the high mis-segregation rate observed due to the high number of faulty attachments in cells with unseparated centrosomes at NEBD [121] may be related to the increase in the proportion of late-aligning polar chromosomes (Figure 5B, part 3). Furthermore, a positive role of spindle elongation during alignment, biorientation, and error correction [21,23,125] implies that the congression that occurs after spindle elongation could be less efficient or more error-prone than the congression that occurs during prometaphase spindle elongation (Figure 5B, part 3). This is supported by the observation that late-aligning polar chromosomes are prone to chromosome mis-segregations in HeLa cells [126]. On the other hand, another study reported that reduced prometaphase spindle elongation decreases the number of lagging chromosomes in human cells [127], probably due to overcharged Aurora B activity, although this phenomenon could also be related to decreased cohesion fatigue. Interestingly, it was demonstrated that unaligned polar chromosomes induce spindle positioning defects in addition to chromosome mis-segregation [128], suggesting a reciprocal link between aberrant spindle positioning and faulty attachments.

As polar chromosomes are characterized by their passage across the polar region, which could adversely impact their alignment compared to other chromosomes, this implies that centrosomes could be important regulators of chromosome alignment. How does the alignment of polar chromosomes occur in human cells without centrioles, the main constituents of centrosomes? Multiple analyzes showed that in cells without centrioles, spindles can bipolarize, align their chromosomes, and continuously divide [90], although at the cost of prolonged spindle assembly, chromosome mis-segregation, DNA damage, cell cycle arrest, and apoptosis [67,129,130]. The increase in the number of mis-segregations could be due to the fact that acentriolar spindles are initially disorganized and later multipolar prior to bipolarization [130], as the multipolar-to-bipolar transition is known to promote merotelic attachments and lagging chromosomes [124]. Interestingly, after centrosome ablation in prophase, cells form functional mitotic spindles but display an increased number of syntelic polar chromosomes at the onset of anaphase [125]. Moreover, centrosome age regulates the propensity of polar chromosomes to get unaligned and mis-segregated through differential regulation of end-on attachment stability in the polar region [26]. In conclusion, centrosomes regulate MT end-on attachments to kinetochores and chromosome congression. Consequently, their correct prepositioning is important for decreasing the proportion of polar chromosomes at NEBD, the number of erroneous attachments, and mis-segregations.

## 5. Regulation of Lateral to End-on Conversion and Error Correction

As it was shown that kinetochores during early prometaphase are mainly captured laterally along the walls of MTs [21,65,96], which sequentially turns into stable end-on attachment [21,131], regulatory mechanisms of lateral to end-on transition have been a hot topic for mitosis researchers. Three main questions emerged: what signaling mechanisms bias for lateral attachment in regions where end-on attachment would be disadvantageous, such as around the spindle pole, what mechanisms ensure the transition to end-on attachment at the appropriate place and time, and what mechanisms ensue when the aforementioned mechanisms fail? First, lateral associations are presumably mediated by the motors that localize to the fibrous corona, cytoplasmic dynein, and CENP-E [132]. Recently, the Ndc80 complex has also been implicated as a component required for lateral attachments during early mitosis in human cells [65,133]. Lateral and end-on attachments are discriminated by molecular markers since only mature end-on attachments recruit components of the Astrin-SKAP complex [96] and release SAC proteins such as Mad1 [22,131,134] (Figure 6).

The Ndc80 complex is essential for the establishment of end-on attachments of kinetochores to MTs [4,113,135]. The Knl1/Mis12 complex/Ncd80 complex (KMN network) is connected to the inner kinetochore primarily by the DNA-interacting CENP-T [136,137,138] (Figure 6). In both yeast and human cells, Aurora B is a crucial factor involved in the control of the lateral to end-on conversion [139,140] (Figure 6, part 1). Aurora B is a component of the CPC, which includes INCENP, Borealin, and Survivin in addition to Aurora B [140]. Interestingly, lateral attachments are not susceptible to MT detachment mediated by the centromeric Aurora B and are not reliant on high interkinetochore tension [140]. Aurora B activity is counteracted by phosphatases recruited by CENP-E, Astrin complex, SKA complex, and KNL1 complex [4,78,96,140,141,142,143,144,145]. The transition to end-on attachment is followed by the loss of outer-kinetochore associated Aurora B, leaning the balance toward the activity of BubR1-associated PP2A-B56 phosphatase, a crucial step for the establishment of end-on attachments [140] (Figure 6, part 2). Further enrichment of the Astrin complex on end-on tethered kinetochores, along with physical separation of centromere-associated Aurora B from outer-kinetochore substrates [146], facilitates the maintenance of mature end-on attachments [140] (Figure 6, part 2). Furthermore, Astrin stabilizes monotelic attachments by opposing attachment destabilization, primarily through its role in delivering PP1 phosphatase to the outer kinetochore (Figure 6, part 2), thus opposing CDK1 kinase activity [147,148], together with coinciding changes in the kinetochore architecture [149]. Thus, in the absence of Astrin, end-on attachments form but are not stably maintained [140,147]. CENP-E and Ska complex also deliver pools of PP1 to the outer kinetochore during the end-on conversion, which counteracts attachment destabilization [142,143,150] (Figure 6, part 2), although their contribution is not essential for the maintenance of end-on attachment in the absence of Aurora B [147]. Concomitantly, the outer-kinetochore checkpoint proteins BubR1, Bub1, and Mps1 that influence attachments and SAC are all removed or reduced after the formation of stable end-on attachment by the dynein-mediated stripping of the fibrous corona [96,131,151].

Erroneous syntelic and merotelic attachments can form during prometaphase [152,153] and need to be corrected to avoid chromosome mis-segregation. However, since lateral attachments dominate during early spindle assembly, it is still unclear where and when erroneous attachments form during early mitotic events in human cells and what the intermediate states are [12,152]. For example, monotelic attachment likely represents a normal transition to amphitelic attachment in the region between the poles [47], while behind the poles, this attachment would be an error that needs to be corrected [125]. Similar to lateral to end-on conversion mechanisms, a key player during error correction is also Aurora B kinase [13,151,154,155]. The main activity of Aurora B in error correction involves the destabilization of incorrect attachment through the phosphorylation of several outer kinetochore proteins that directly bind to MTs, including components of the KMN network [55,146,153,156,157,158,159] (Figure 6, parts 3 and 4).

How does Aurora B discriminate between correct and incorrect attachments? The most appealing model is based on the low tension of erroneous attachments when compared to amphitelic ones [146,159,160]. Such spatial positioning models of the Aurora B function are based on a physical distance between the kinase and its kinetochore substrates, either by a diffusible phosphorylation gradient or by Aurora B being positioned on a long tether [151]. However, low interkinetochore distances do not induce Aurora B-mediated error correction [161], decreased phosphorylation of incorrect attachments can ensue, regardless of their tension state [162], and amphitelic attachments can form in cells with a kinetochore-proximal pool of Aurora B if cohesion is stabilized [163,164]. Therefore, it was suggested that the inner centromere localization of Aurora B is not a prerequisite for the phosphorylation of erroneous kinetochore-MT attachments nor for the stabilization of correct attachments [159], although it is crucial for the regulation of MCAK activity to destabilize MTs [55,165]. Recently, distinct populations of Aurora B were found localized to the inner centromere, outer centromere, and outer kinetochore, although the receptor for Aurora B at the outer kinetochore is unknown [149]. While centromeric Aurora B appears necessary for error correction, intriguingly, it is not required for phosphorylation of kinetochore substrates, which presumably depends only on Aurora B localized to the kinetochore [163,166,167,168]. Once correct kinetochore-MT attachments are formed, and tension is established, the Aurora B pool at kinetochores is presumably lost [149]. Further mechanisms that could discriminate different attachments include progressive restriction of attachment geometry [149,169] and MT-pulling or tension-associated active detachment [170].

How exactly does Aurora B mediate the destabilization of incorrect attachments? Aurora B phosphorylation could promote both detachment and depolymerization of end-on attached MTs [56,135,153,158] (Figure 6). Recent experiments indicate that Aurora B promotes detachment under high tension and depolymerization under low tension [156]. Therefore, tension is probably not an input for error correction, but it regulates downstream response to Aurora B phosphorylation. This could be important in distinguishing low-tension syntelic attachments from merotelic attachments that are usually stretched [55,146] (Figure 6). The major unresolved question is how Aurora B leaves the correct attachments unaffected but corrects the stretched merotelic attachments. This could be because at a merotelic kinetochore, only the attachments of MTs emanating from the distal pole are presumably within the centromeric Aurora B phosphorylation gradient and will be destabilized, whereas the correct attachments will be unaffected (Figure 6, part 4). Moreover, a recent model proposed that MTs from the distal pole provide a path for Aurora B diffusion because they pass close to the centromere; thus, the active kinase grasps only erroneous attachments without affecting the ends of MTs from the proximal pole [162] (Figure 6, part 4). Regarding Aurora B-mediated MT depolymerization that brings kinetochores with syntelic attachment to spindle pole [56], it was proposed that this could be followed by MT detachment mediated by pole-localized Aurora A that also phosphorylates the Ndc80 complex [54,171] (Figure 6, part 3). Although Aurora A is mostly localized around spindle poles [155], it has recently been connected even with error correction of aligned chromosomes promoted by chromosome oscillations [172].

In addition to their role in the regulation of the stability and dynamics of MTs, Aurora kinases are also involved in the regulation of the main kinetochore motors involved in the congression. For example, phosphorylation of CENP-E near its motor domain by Aurora A and Aurora B is necessary for the congression of polar chromosomes [142] (Figure 6, part 1). Interestingly, phosphorylation of CENP-E by Aurora kinases reduces its affinity for MT, and it is not yet clear how reducing MT affinity would increase the processivity of CENP-E required for polar chromosome congression. Dynein, on the other hand, is regulated by Plk1 phosphorylation in a chromosome position-dependent manner [173,174] but is also indirectly controlled by Aurora B [175,176]. Interestingly, CENP-E is known to counteract Aurora-B mediated Ncd80 phosphorylation, as inhibition of CENP-E increases the phosphorylation of Ncd80 on polar chromosomes in a tension-independent manner [177]. Furthermore, Aurora B inhibition induces the removal of Mad2 from polar chromosomes after CENP-E depletion [18,103], suggesting that CENP-E could oppose the effects of Aurora B on MT destabilization, possibly by its role in PP1 delivery. Clearly, more work is needed to unravel the complex coordination between Aurora kinases and the main motor proteins involved in the congression of polar chromosomes, especially in different spindle regions and at different time points during early mitosis.

## 6. Mis-Segregation and Aneuploidy of Polar Chromosomes

### 6.1. Different Ways to Mis-Segregation through Polar Chromosomes

There are multiple possible pathways by which unaligned polar chromosomes can lead to mis-segregation and aneuploidy (Figure 7). First, a polar chromosome can remain unaligned through metaphase and anaphase and mis-segregate while still stuck at the polar region (Figure 7, route 1). Interestingly, the frequencies of unaligned chromosomes in anaphase across different cell types are not known, although they are expected to be low [124,178]. This is because cells have intricate systems to avoid the occurrence of unaligned chromosomes, but if such chromosomes persist, they must pass the control of SAC [11] (Figure 7, route 1). Although SAC activity can be weakened, which would allow mis-segregation of unaligned polar chromosomes by precocious anaphase start [179], from extensive data, it is becoming clear that SAC-related deficiencies are rare in human tumors, and even cells with high CIN generally do not enter anaphase precociously [180,181]. However, attenuation of SAC promotes aneuploidy [182], weakened SAC is associated with certain aneuploidies during early embryogenesis [183], and rare genetic disorders with altered SAC, such as mosaic variegated aneuploidy (MVA), are documented [184]. Although the level of SAC response is understudied in unperturbed systems, much more is known about the SAC response after different perturbations. For example, several studies have revealed that SAC is not an all-or-nothing response in human cells, but it scales with the number of unattached kinetochores [185,186], delaying anaphase onset for up to a few hours. Unaligned chromosomes produced similar mitotic delays in Ptk1 cells [187]. These findings imply that only a low number of unattached polar chromosomes could induce moderate mitotic delays, which would lead to mis-segregation of unaligned chromosomes.

Accordingly, studies that used CENP-E perturbations reported mis-segregation of one or a few unaligned polar chromosomes after moderate mitotic delays in HeLa [103], RPE1 [188,189], and mouse embryonic cells [190]. A similar phenomenon was observed after treatment of cells with nanomolar doses of nocodazole [186]. Interestingly, the SAC response was robust since Mad1 was recruited on unaligned kinetochores in such cells, and mitotic delays were at the level of a few hours. Moreover, anaphase onset was not the result of ‘cohesion fatigue’ or ‘mitotic slippage’ [191,192] since it was preceded by the loss of Mad proteins from unaligned kinetochores, implying that both sister kinetochores acquired end-on connections that were able to satisfy the SAC [103,186,188,189]. However, it is unclear what types of attachment could satisfy the SAC at the pole since syntelic, monotelic, and lateral attachments induce a large SAC response [134], although it is possible that the SAC signal wears faster over time if stable end-on attachments are present at one of the sister kinetochores [125,193]. A recent study presented an intriguing model in which difficulties associated with the inability of unaligned chromosomes to congress are not necessarily caused by defects related to MTs, SAC, or motors but rather due to the presence of a complex system of organelle remnants behind the spindle poles, termed endomembranes, which could ensheat polar chromosomes and prevent their efficient capture by MTs, resulting in aneuploidy [189]. However, for definitive conclusions about the mechanisms underlying unaligned chromosome mis-segregation in unperturbed cells, one would need to track the origin and fate of mis-segregations through whole mitosis, similar to what was done recently from metaphase to telophase in human cells [194].

Mis-segregation of unaligned chromosomes could induce extra chromosomes in the main nuclei or the formation of micronuclei in one of the daughter cells [179,195] (Figure 7, route 1), where the latter is particularly detrimental to genome stability as it is related to chromothripsis [196]. Interestingly, the micronucleus generated from the polar chromosome is different from the one generated by a lagging chromosome stuck in the cleavage furrow, as nuclear lamina defects and the response to DNA damage are not pronounced in micronuclei that originate from polar chromosomes [195] (Figure 7). This phenomenon could drastically affect the propensity to propagate unaligned chromosomes over generations. However, a recent study indicated that micronuclear stability is determined more by the identity and length of chromosomes trapped within the micronucleus than by their mis-segregation position [178]. Regardless of the mechanisms, if an unaligned polar chromosome satisfies the SAC and results in aneuploidy, it is expected that daughter cells would struggle in proliferation independently of aneuploidy, as prolonged prometaphase activates p53-p21 dependent apoptotic response that blocks further daughter cell proliferation [197,198]. The monosomic daughter cell would be at an even higher risk of cell cycle arrest since, in human somatic cells, chromosome loss impairs proliferation and genomic stability more than chromosome trisomy [199]. Recently, it was shown that even shortened mitosis induced upon SAC inhibition increased the apoptotic response in human cells [200]. Overall, we speculate that among the mis-segregations caused by polar chromosomes, selection would favor those that are able to satisfy the SAC in the well-defined time frame (Figure 7).

Second, even if the polar chromosomes align at the metaphase plate, there is still a possibility that their attachments could be erroneous. The most studied type of erroneous attachment is the merotelic attachment [12] (Figure 7, route 2). As kinetochores in late prometaphase and metaphase cells are oriented parallel to the main spindle axis [21,22], this back-to-back orientation minimizes the chance that the kinetochore captures MTs from the distant pole [125]. This is, however, not the case with polar chromosomes, as during early prometaphase, polar chromosomes are characterized by random orientations with respect to the main spindle axis [21]. Thus, are polar chromosomes more susceptible to merotelic attachments? There are prominent mechanisms that ensure kinetochores avoid stable attachment near the pole and mechanisms that correct such attachments if they occur (see Section 4.3 and Section 5). However, because of their proximity to the pole, it is conceivable that polar chromosomes could have a higher risk of acquiring syntelic attachments, which may convert to merotelic, although such a hypothesis requires experimental testing. Interestingly, in a photoactivation study in HeLa cells, it was reported that late-aligning polar chromosomes increase the rate of lagging chromosomes [126], and lagging chromosomes are mostly associated with merotelic attachments [152].

If merotelic attachments are formed and persist until anaphase onset, this does not necessarily imply aneuploidy. In a set of recent studies, it was reported that the Aurora B midzone gradient mediates phosphorylation of outer kinetochore proteins even during anaphase [201], at similar sites as in pre-anaphase cells [158]. Thus, it seems that error correction mediated by Aurora B has an additional layer operating in early anaphase [194,202] (Figure 7, route 2), which could explain previous observations that the proportion of lagging chromosomes during anaphase is by an order of magnitude higher than the proportion of cells with aneuploidy in the same population [203]. Furthermore, this led to a redefinition of lagging kinetochores by introducing a new term of ‘lazy’ kinetochores, transiently lagging kinetochores that are quickly and efficiently corrected during the early anaphase by the Aurora B-dependent mechanism [194] (Figure 7, route 2). However, in certain situations, even moderate alignment defects can induce malfunctioning nuclear morphologies and micronuclei [204], calling for long-term tracking of chromosome fates during mitosis.

Third, even if polar chromosomes align at the metaphase plate, they could be characterized by various types of alignment problems compared to other chromosomes. For example, late-aligning polar chromosomes could have perturbed end-on attachment stability because global MT stability increases as the cell progresses from prometaphase to metaphase [205]. As out-of-plate movements are extremely rarely observed in human non-transformed cells [21], instability of position within the metaphase plate could be particularly pronounced in tumor cells due to the defective stability of kinetochore MTs (Figure 7, route 3), a characteristic trait of most chromosomally unstable tumors [206,207]. Tumor systems are expected to have a larger fraction of polar chromosomes at NEBD [18,126,188]. Overall, we speculate that the stability of chromosomes within the plate may depend on the time of chromosome arrival at the equator. Instability of chromosome position within the plate would put a burden on the mitotic fidelity by increasing the number of unaligned chromosomes, thus triggering a loop of defects associated with these chromosomes (Figure 7, route 3). In this feedback loop, late-aligning chromosomes are frequently expelled from the pseudo-metaphase plate back to the poles, where they must avoid both formations of merotelic attachments and premature satisfaction of SAC (Figure 7). Consistent with this assumption, unaligned chromosomes are frequently observed in systems that are characterized by a higher rate of lagging chromosomes and defective stability of MT, such as immature mouse organoids [208] and human cells treated with nanomolar doses of nocodazole [22,186]. Furthermore, unaligned chromosomes are not efficiently phosphorylated by Aurora B in tumor cells, contrary to non-transformed cells [157], which could explain their inefficient correction in tumor systems.

Altogether, polar chromosomes may be prone to various types of mis-segregations and alignment problems that could deleteriously affect mitotic fidelity, especially in tumor cells with perturbed MT stability and error correction mechanisms. However, it is currently unclear if chromosome mis-segregations and whole-chromosome aneuploidies are biased and to what extent toward certain chromosomes, with much evidence supporting [188,209,210,211,212] and some discouraging [213] such a hypothesis in vertebrate model systems after different spindle perturbations. The reported biases are related to the size of the chromosome or kinetochore [178,209,210], the duration of prometaphase and cohesion fatigue [209], or the level of centromere-associated proteins [211,212].

### 6.2. Polar Chromosomes in Cancer—Aneuploidy and CIN

Aneuploidy is a hallmark of human cancers and is tightly interlinked with high rates of chromosome mis-segregation and CIN [14]. Chromosome congression and alignment defects are reported in human tumors with high CIN rates [7,181], although the relevance of such observations for tumorigenesis is unclear, as there are many genetic disruptions that could lead to this phenotype. Current evidence does not support the conclusion that chromosome congression defects can drive tumor formation in fly models [214], while in mouse CIN models, a low level of CIN caused by mis-segregation of unaligned chromosomes could lead to tissue-dependent transformation [190]. Elevated CIN rates after SAC perturbations, which are often associated with a large number of unaligned chromosomes, can prevent [215] or promote [214,216] tumor formation in mouse and fly models. Thus, our current understanding of the connections between alignment or SAC defects and tumorigenesis is incomplete.

Regarding human cancers, different molecular elements involved in congression are altered at the genetic or protein level in certain cancers that often correlate with tumor grade and progression, including CENP-E overexpression or downregulation, Kif18a overexpression, Kif4a downregulation or overexpression [217,218,219,220,221], among others [7]. However, although, in principle, it is possible that deregulation of kinesin-related proteins involved in congression is directly tied with aneuploidy generation and tumor development, it could be related to other functions of these proteins or could be a secondary effect obtained through tumor evolution. Finally, more direct studies of tumor systems reported involvement of unaligned chromosomes in certain types of tumors. For example, unaligned chromosomes are common in high-grade serous ovarian carcinoma cells [222] and to certain extent in colorectal cancer cells [223]. In both of these tumors, unaligned chromosomes are associated with markedly elevated MT assembly rates tied to overreactive Aurora A, and in colorectal cancer, also to the deregulated BRCA1-Chk2 signaling axis [222,223]. More recently, it was shown that phosphorylation of BRCA2, which is often mutated in breast cancer patients, by Plk1 is important for the formation of a complex between BRCA2, BubR1, and the phosphatase PP2A. A defect in this function of BRCA2 manifests itself in chromosome misalignment and unalignment, chromosome mis-segregation, mitotic delays, and aneuploidy, eventually leading to CIN [224]. Lastly, unaligned chromosomes induced by Mad2 overexpression in the mammary glands resulted in extensive CIN, which led to the generation of abnormal cells that survived the strong selective pressure of oncogene withdrawal [225]. In agreement with these results, CIN generated by unaligned chromosomes resulted in the increased genetic diversity of cancer cells, which could evade oncogene addiction [226]. There are probably additional examples of unaligned chromosomes occurring in certain tumor types, as mis-segregations are prevalent in human tumors [181], but to establish direct and clear connections between the two, it would be essential to live-image cells from different tumor types throughout mitosis at high spatiotemporal resolution.

## 7. Conclusions and Future Perspectives

In conclusion, polar chromosomes show a set of broad biological behaviors that separate them phenotypically from other chromosomes in human cells. Their specific features include the requirement for kinetochore motor proteins to move poleward and to congress, passage across the large polar region, passage close to the pole where the chance of acquiring erroneous attachments is higher and where end-on attachments are actively destabilized, and reliance on precise spatial and temporal control of the lateral to end-on conversion. Such peculiarity could, in principle, make peripheral polar chromosomes more prone to late-alignment, unalignment, or merotelic attachments during mitosis, especially in tumor systems that are characterized by hyper-stable MT attachments and deficient error correction mechanisms. Therefore, conditions that elevate the number of chromosomes on this alignment route, such as the prometaphase pathway of spindle assembly that frequently operates in error-prone tumor cells, could put a large burden on cellular systems that ensure error-free mitosis, mainly SAC and error correction. As error correction is often flawed in tumors, and SAC can delay mitosis only for a limited time, more erroneous attachments can persist into anaphase in cells when under a heavy burden, thereby fueling error-prone mitosis. However, much more work is needed to establish clear connections between the observed regularities of spatial pathways operating during prometaphase for different chromosomes and their tendencies to mis-segregate. In addition, complex molecular signaling mechanisms that operate during congression, error correction, and implications of those on the formation and correction of different erroneous attachments await more comprehensive studies. An important part of such studies would be long-term tracking of chromosome fates during whole mitosis in different systems, including non-transformed and tumor cells, which would lead toward achieving the goal of discriminating the cellular consequences of individual whole chromosome aneuploidies from different sources [227]. Finally, it will be important to establish whether defects in chromosome alignment and the underlying molecular mechanisms are directly responsible for human diseases such as cancer and whether targeting chromosome congression represents an effective therapeutic approach. However, such a massive effort would require a substantial improvement in cooperation between cell biologists studying the mechanisms of mitosis and cancer researchers working on the mechanisms of tumorigenesis.

## Figures and Tables

**Figure 1 cells-11-01531-f001:**
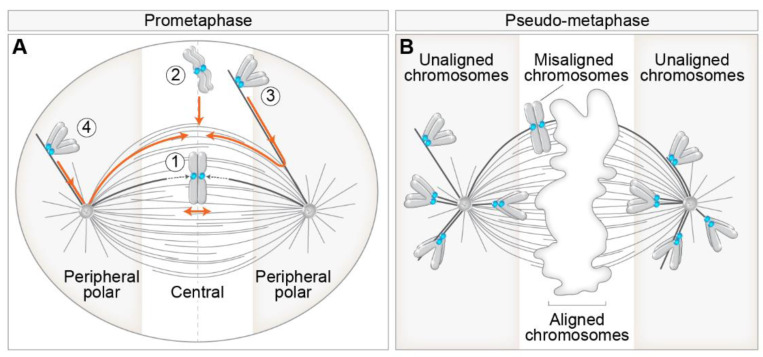
Different routes to chromosome alignment based on the initial position of chromosomes with respect to spindle poles and consequences of aberrant alignment during pseudo-metaphase. (**A**) Initial position of chromosomes with respect to spindle poles defines motor-dependent (peripheral polar) and independent (central) regions important for chromosome alignment during early prometaphase. Different pathways to chromosome alignment to spindle midplane, noted with a circled number, are depicted in orange color. Peripheral polar and central regions are indicated by grey and white color, respectively. Centrosomes are labeled as circles with two centrioles inside. The dotted lines within the spindle indicate microtubule growth. (**B**) Definition of aligned, misaligned, and unaligned chromosomes in the pseudo-metaphase spindle. The white area in the middle of the pseudo-metaphase spindle denotes chromosomes aligned within the metaphase plate. The boundaries between the peripheral polar and central regions signify the dependence on kinetochore motors for alignment. In all figures, please see the text for details and references.

**Figure 2 cells-11-01531-f002:**
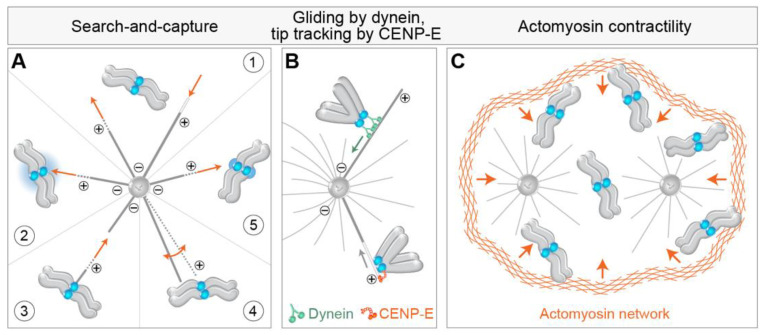
Mechanisms of kinetochore search-and-capture and gathering on the spindle. (**A**) (1) Chromosome alignment involves the microtubule search for kinetochores (blue circles) by dynamic growth (dotted lines and orange arrow pointing away from the microtubule tip) and shrinkage (empty white line and orange arrow pointing to the microtubule tip). The efficiency of search-and-capture is facilitated by different additional mechanisms, including (2) biased microtubule growth towards the kinetochore via the Ran-GTP gradient (blue gradient), (3) microtubule growth from the kinetochore, (4) microtubule pivoting (orange curved arrow) around the spindle pole, and (5) kinetochore expansion before microtubule capture. Microtubule plus and minus ends are denoted by encircled + and − signs. The spindle is represented as monopolar for simplicity. (**B**) The successful capture event for the polar chromosome is followed by the gliding of the kinetochore laterally along the microtubules mediated by dynein walking toward the minus end of microtubule (top) or by kinetochore tracking of the depolymerizing microtubule tip (empty white line) by CENP-E. The green arrow represents the direction of dynein walking, and the grey arrow represents the direction of microtubule depolymerization. (**C**) Chromosome gathering on the spindle is also facilitated by the synchronous actomyosin contractility (illustrated with orange arrows) during early prometaphase, independently of microtubules.

**Figure 3 cells-11-01531-f003:**
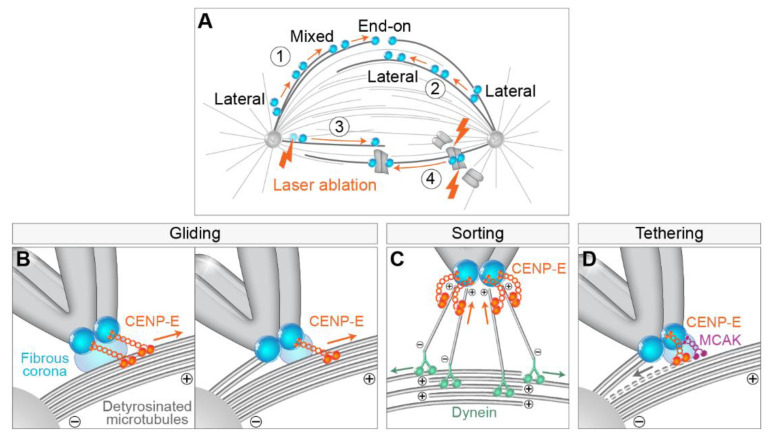
Main features and models of CENP-E-mediated chromosome congression to the spindle midplane. (**A**) (1) Congression of polar chromosomes to the equator involves the transition of lateral to end-on attachments of kinetochores (blue circles) to microtubules, (2) although congression is achievable by lateral-only interactions. Note that the sister kinetochores gradually change orientation to be parallel to the main spindle axis, and the interkinetochore distance gradually increases during congression in (1) and (2). (3) Both sister kinetochores and (4) chromosome arms are not prerequisites for congression. Orange arrows indicate congression movement over time. Lightning signs denote laser ablation of a specific structure. (**B**) Gliding model: Congression is facilitated by the CENP-E plus end directed walking and pulling laterally attached kinetochores (left) or mixed laterally/end-on attached sister kinetochores (right) toward microtubule plus end. Part of the centrosome is represented by the gray semicircle in the bottom left. (**C**) Sorting model: By walking towards the plus end, CENP-E sorts small kinetochore-nucleated microtubules in a way that their plus ends are oriented toward the kinetochore, while minus ends are connected to dynein, which facilitates their poleward-directed growth by walking along antiparallel microtubules toward their minus ends. (**D**) Tethering model: As the first step in end-on conversion, CENP-E is involved in the tethering of kinetochores to lateral surfaces of microtubules, while in the second step, MCAK is involved in the depolymerization (grey dashed line) and resolving of lateral microtubule attachments to kinetochores. The orange and green arrows in (**B**–**D**) represent the direction of motor movement, and the gray arrow represents the direction of microtubule depolymerization.

**Figure 4 cells-11-01531-f004:**
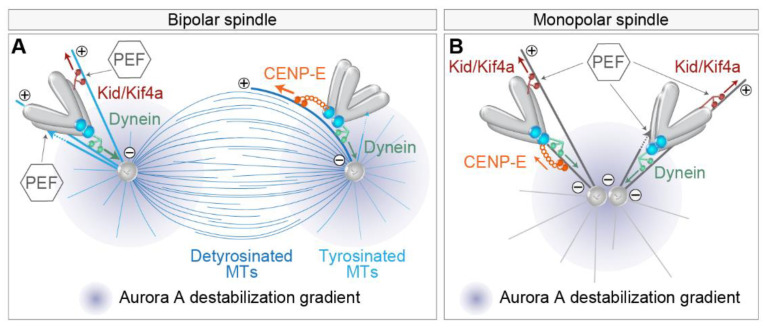
Mechanisms that set the distance between the chromosome and the spindle pole in bipolar and monopolar spindles. (**A**) Chromosome proximity to the pole where the Aurora A gradient is centered (purple gradient) controls its end-on attachment status to microtubules by the balance of poleward dynein (green arrows), anti-poleward polar ejection (PEF), and CENP-E (orange arrows) forces in the bipolar spindle. The extent of CENP-E motor activity behind and in the front of the spindle pole is controlled by the tubulin tyrosination state (microtubules in two different shades of blue). PEF activity is represented by the plus end directed walking of Kid and Kif4a motors on microtubules (red arrows) or direct microtubule pushing into the chromosome arm by polymerization (dotted lines from a microtubule tip). The thickness of the colored arrows indicates the strength of each force. MTs, microtubules. (**B**) Mechanisms that regulate the proximity of chromosomes to poles in the monopolar spindle. Legend as in (**A**).

**Figure 5 cells-11-01531-f005:**
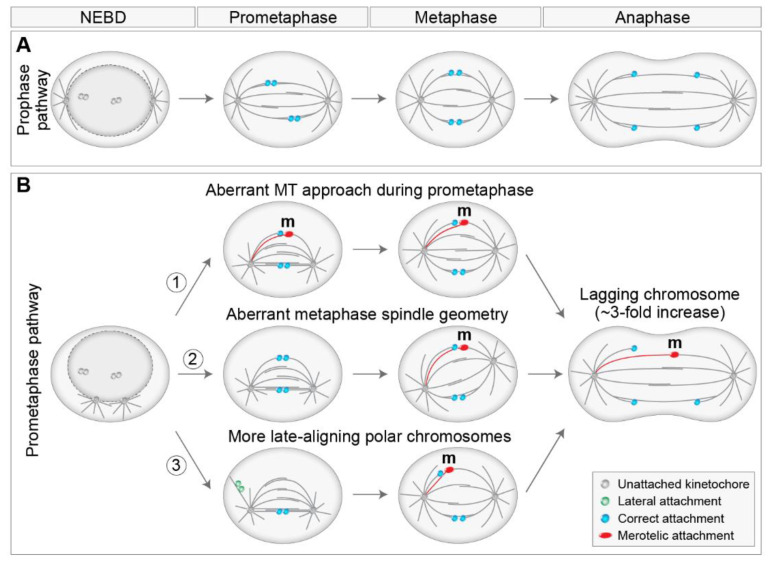
Models by which the degree of centrosome separation upon nuclear envelope breakdown (NEBD) influences the fidelity of chromosome segregation. (**A**) Two spindle poles separate before NEBD, termed the prophase pathway. This pathway is not characterized by chromosome mis-segregation. (**B**) Spindle poles separate after NEBD, termed the prometaphase pathway. The prometaphase pathway is associated with a three-fold increase in the number of lagging chromosomes compared to the prophase pathway through different mechanisms that promote erroneous merotelic attachments (1)–(3). An erroneous microtubule from the distal pole connected to a merotelic kinetochore is labeled red. The arrows indicate changes over time. Kinetochores with different attachment types are color labeled according to the legend on the bottom right. MT, microtubule; m, merotelic attachment.

**Figure 6 cells-11-01531-f006:**
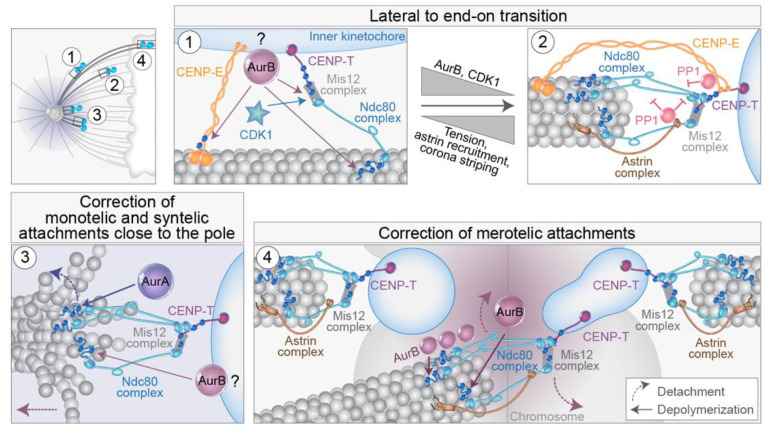
Mechanisms that control the lateral to end-on transition and error correction in different spindle regions. Different attachment types from the scheme at the top left are enlarged in panels (1)–(4). (1) and (2) Simplified view of the main molecular machinery that regulates the transition from lateral (1) to end-on (2) attachment of kinetochores to microtubules. The spindle assembly checkpoint is turned on in (1) and turned off in (2). A pool of Aurora B is present at the outer kinetochore, although the kinetochore receptor for this pool is unknown (“?”). (3) Simplified view of the main molecular machinery that regulates the correction of monotelic and syntelic attachments close to the spindle pole in the gradient of Aurora A activity (same color as the circled AurA), and (4) the correction of stretched merotelic attachments in the gradient of centromeric Aurora B activity (same color as the circled AurB) or by the Aurora B diffusing along the microtubule that originates from the distant pole (small Aurora B circles). Arrows from Aurora kinases denote phosphorylation of various kinetochore targets. Dashed arrows from kinetochore targets denote either detachment (curved arrows) or depolymerization (straight arrows) (legend in part 4). Small blue dots represent phosphorylated residues. AurA, Aurora kinase A; AurB, Aurora kinase B.

**Figure 7 cells-11-01531-f007:**
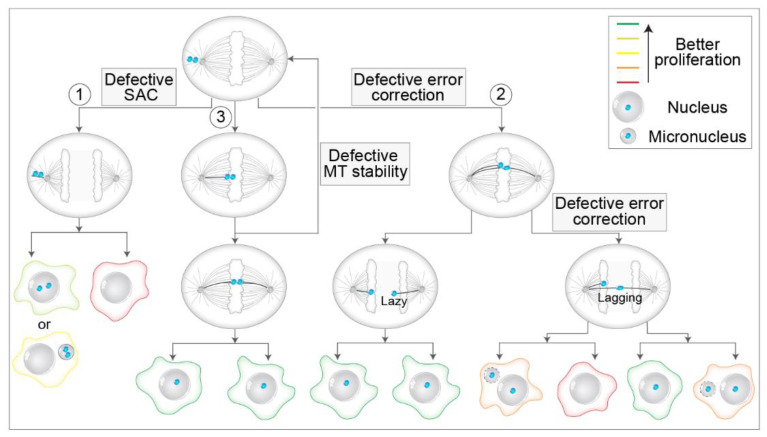
Routes to chromosome mis-segregation through unaligned polar chromosomes and their consequences. A process flow diagram depicting different pathways of the pseudo-metaphase cell with an unaligned polar kinetochore pair and their consequences: (1) mis-segregation of the unaligned chromosome by precocious anaphase start, (2) alignment to the metaphase plate (white area) with the merotelic attachment, which can be (left path) either resolved by error correction and result only in lazy kinetochore with appropriate attachment or (right path) left unresolved resulting in the persistent lagging chromosome with merotelic attachment, and (3) alignment of the polar kinetochore to the metaphase plate which can result in either successful biorientation (bottom path) or return of the kinetochore pair back to the pole and the initial position due to the defective stability of kinetochore microtubules (right path). The arrows signify the changes in time. Multiple arrows starting from one cell signify the opposite result of a process. Red to green lines on the cell borders represent the level of detrimental effects on the proliferative capacity of daughter cells (legend on the top right). Large grey spheres depict nuclei, and small grey spheres micronuclei (legend on the top right). Micronuclear membrane instability is depicted by dashed lines.

## Data Availability

Not applicable.

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
