# Peer review of "Polar Chromosomes—Challenges of a Risky Path"

_cells, 2022, doi:10.3390/cells11091531_

Round 1
Reviewer 1 Report
the paper is an extensive analysis of past and recent literature on the mechanisms of chromosome congression, with specific emphasis on polar chromosomes. Thus, it is a valuable paper for those interested in the subject.
However, the large number of different mechanisms that have been reviewed ( from chromosome movement to error correction to the consequences of miss-segregation) does not allow a simple readibility of the paper.
I see two main points that need to be worked out: 1 is to distinguish between the analysis of a general mechanisms (e.g. search and capture or chromosome oscillations) and the specific subpathway directed to polar chromosomes. For example introducing first the mechanism in general and then moving to polar chromosomes.
The most critical point to be addressed are the figures. These do not represent a way of clarifying the text but are very confused and confusing. They need to be heavily revised, since they are often incoherent in size, dimension or icons. They need to be semplified and to have explicatory legends that are now not clear enough.
fig. 1B the white area representing aligned chromosomes should be changed to be more realistic.
Fig.2 The search and capture should describe also chromosomes within the spindle region and it is a general mechanism tha brings to chromosome congression. ity sshould be introduced its entirety including cenp_E role, before to discuss polar chromosomes. Figure 2 C can be deleted.
Figure 3. The MT bundle should be in proportion to centrosome (much bigger) and kinetochore. Fig.3 C is not described and it is very confused and possibly wrongly referred in the text
Fig. 4 Chromosome arm also in the chromosome inside the spindle are subjected to PEF that position arm outside the spindle.
FIG5. also does not help the reader very much. The colors of kinetochores are confounding, fewer MTs would help. In part B the arrow should signify changes with time but they do not help.
Also in this case an introduction on attachments in general and merotelic before going into the main subject would help.
Fig. 6 must be changed completely The centromere as an halo is misleading. Aurora B as a circle positioned nowhere or sac as a mechanism there, are all nonsense. Less information but more clear-
Fig. 7 Arrow point to time or opposite results. error correction must be alternative to lack of error correction, defective MT stability going back? START?
A point to be changed is that the authors cite NEBD as the time of cell shape changes while that is a prophase mechanism (first page and below). Also the mitotic spindle does not inityiate at NEBD (line 12).
Last but extremely important aneuplody is not a deviation from a multiple of a haploid!
The sentence although .. line 4 is not clear.
Author Response
Rebuttal letter
We thank the referees for their insightful comments. In the rebuttal letter we have marked the new text in blue.
Reviewer #1:
1) The paper is an extensive analysis of past and recent literature on the mechanisms of chromosome congression, with specific emphasis on polar chromosomes. Thus, it is a valuable paper for those interested in the subject.
Authors: We thank the reviewer for the comments and extensive examination of the manuscript.
However, the large number of different mechanisms that have been reviewed (from chromosome movement to error correction to the consequences of miss-segregation) does not allow a simple readability of the paper.
Authors: We understand the manuscript is covering a large set of mitotic mechanisms, but as this is to our knowledge the first review manuscript covering specifically polar chromosome behavior, and its implication for mitotic fidelity, we covered all important facets of spindle biology connected to polar chromosomes. In the revised manuscript, we have simplified all chapters, and reorganized the paragraphs to better reflect the main points, and furthermore cited appropriate review papers for a more general and global review of mechanisms covering all aspect of chromosome behavior.
I see two main points that need to be worked out: 1 is to distinguish between the analysis of a general mechanisms (e.g., search and capture or chromosome oscillations) and the specific subpathway directed to polar chromosomes. For example, introducing first the mechanism in general and then moving to polar chromosomes.
Authors: We thank the reviewer for the suggestions. We have reorganized parts of the text to better reflect general mechanisms of alignment applicable to all chromosomes. We have introduced new chapter 3. “General models of chromosome alignment” where we discuss the basic and supplementary mechanisms of search-and-capture (section 3.1. “Search-and-Capture model”). We have also referred the reader toward a recent review article on the issue of search-and-capture, and briefly introduced all currently dominant models of maintenance of chromosome alignment within the equatorial plane of the spindle (section 3.2. “Oscillating at the equator - maintenance of alignment”). Additionally, we have now pointed out through the rest of the article when a certain mechanism is generally applicable to all chromosomes.
The most critical point to be addressed are the figures. These do not represent a way of clarifying the text but are very confused and confusing. They need to be heavily revised, since they are often incoherent in size, dimension or icons. They need to be simplified and to have explicatory legends that are now not clear enough.
Authors: We thank the reviewer for the comment. We have now levelled the size and dimensions of figures, and solved problems with inconsistencies through figures, as per reviewer specific comments. We have also simplified certain figures to better represent the text and the take-home message, and rewritten substantially the figure captions to be more detailed and informative about specific figure parts.
fig. 1B the white area representing aligned chromosomes should be changed to be more realistic.
Authors: We thank the reviewer for the observation. We have redrawn and scaled the metaphase plate representing aligned chromosomes based on a live-cell image of the metaphase spindle from our original scientific paper to be more realistic.
Fig.2 The search and capture should describe also chromosomes within the spindle region and it is a general mechanism that brings to chromosome congression. It should be introduced its entirety including Cenp-E role, before to discuss polar chromosomes. Figure 2C can be deleted.
Authors: We thank the reviewer for the comment. To simplify the main massage and avoid possible confusion we have redrawn this part of Figure 2 as a simple monopole depicting basic model of search-and-capture together with additional mechanisms as separated subpanels of Figure 2A. Regarding the text part of the manuscript, we have reorganized the manuscript to better reflect the general mechanisms of alignment applicable to all chromosomes. Although CENP-E can have effect on aligned chromosomes, it was shown that it is crucial only for alignment of peripheral polar chromosomes (Barisic et al., 2014, Nature Cell Biology), hence we favored the organization of text that asserts CENP-E role in alignment of polar chromosomes. We left Figure 2C as part of Figure 2, as this is an important part of gathering of peripheral chromosomes on the spindle (Booth et al., 2019, eLife).
Figure 3. The MT bundle should be in proportion to centrosome (much bigger) and kinetochore. Fig.3 C is not described and it is very confused and possibly wrongly referred in the text.
Authors: We thank the reviewer for these helpful observations. We have redrawn Figure 2 to better reflect dimension of centrosome, MT bundle and kinetochore by drawing centrosomes as a much larger structure, and kinetochores more in scale with the width of the microtubule bundle. As Figure 3C was indeed confusing and incorrectly drawn, we have redrawn it to be correct by introducing antiparallel microtubules, dynein walking towards minus end on antiparallel microtubules, and both kinetochores nucleating microtubules, now representing a seed of a future amphitelic attachment, as proposed by the presented model. We have also rewritten the text, and introduced new sentences to better reflect general implications (“Also, because of synchronous bi-orientation of kinetochores in human cells during prometaphase [23], the model of kinetochore-nucleated MTs proposes that the process of chromosome biorientation is more deterministic than is usually assumed [17].”) and the possible role of the augmin complex (“Amplification of short and sorted kinetochore-nucleated MTs could be aided by the augmin complex [93] that contributes to kinetochore MT growth even in the absence of pre-existing centrosomal MTs [94].”) in the presented model.
Fig. 4 Chromosome arm also in the chromosome inside the spindle are subjected to PEF that position arm outside the spindle.
Authors: We thank the reviewer for this observation. To better reflect the role of PEFs in positioning the chromosome arm outside the spindle the Figure 4A is now redrawn by placing the chromosome arms outside the spindle area in a characteristic “V” shape.
FIG5. also does not help the reader very much. The colors of kinetochores are confounding, fewer MTs would help. In part B the arrow should signify changes with time but they do not help.
Authors: We have simplified this figure by omitting the original kinetochore color code and introducing a much simpler color code defined by the newly introduced legend, and by reducing the number of microtubules within the spindle, as per reviewer suggestions.
Also, in this case an introduction on attachments in general and merotelic before going into the main subject would help.
Authors: We thank the reviewer for this suggestion. We have introduced a new sentence in Introduction part of the manuscript, close to the definition of amphitelic attachment, that defines main types of erroneous attachments in the spindle (“The main erroneous attachments include syntelic attachment, where both sister kineto-chores interact with MTs emanating from the same spindle pole, and merotelic attachment, where a single kinetochore is connected to both spindle poles [12].“).
Fig. 6 must be changed completely. The centromere as an halo is misleading. Aurora B as a circle positioned nowhere or sac as a mechanism there, are all nonsense. Less information but more clear.
Authors: We have drawn part of the chromosome on Figure 6 (part 4) to make it clear that the gradient of Aurora B is positioned on the centromere. We have significantly rewritten the caption to clarify all parts, and introduced a legend in the figure explaining detachment and depolymerization parts that are result of activity of Aurora kinases. We have positioned Aurora B more realistically and introduced new sentences in the text and figure caption to better reflect the notion that outer-kinetochore adaptor of Aurora B is not known which makes it difficult to place the kinase close to some defined complex. Text: “Recently, distinct populations of Aurora B were found localized to the inner centromere, outer centromere, and the outer kinetochore, although the receptor for Aurora B at the outer kinetochore is unknown [149].”. Figure caption: “A pool of Aurora B is present at the outer kinetochore, although the kinetochore receptor for this pool is unknown (“?”).”. We have also removed the SAC as a mechanism from the figure, and instead introduced a sentence in the caption to reflect the notion when the SAC is turned on or off depending on the status of kinetochore end-on attachment (“Spindle assembly checkpoint is turned on in (1) and turned off in (2).”).
Fig. 7 Arrow point to time or opposite results. error correction must be alternative to lack of error correction, defective MT stability going back? START?
Authors: We thank the reviewer for the comments. We have redrawn this figure to be more simple and clearer by placing subsequent time points in a process flow diagram in a vertical order. We have also removed the START sign. We have also rewritten error correction and SAC parts to better reflect their putative roles by changing the concept to one where their defects are fueling certain decision making in the branches of flow diagram. Defective MT stability part was also redrawn to be more simple and clearer. We have also modified the arrows, and clarified the meaning of arrows in the figure caption: “The arrows signify the changes in time. Multiple arrows starting from one cell signify opposite result of a process.”.
A point to be changed is that the authors cite NEBD as the time of cell shape changes while that is a prophase mechanism (first page and below). Also, the mitotic spindle does not initiate at NEBD (line 12).
Authors: We have rewritten that part to better reflect time-dependent nature of events occurring during early mitosis: “During early prometaphase, the cell shape changes, the interphase array of MTs is reorganized, and MT dynamics drastically increases as MTs invade an opened nuclear space packed with chromosomes [6]. This initiates interaction between MTs, and MTs and kinetochores, resulting in the formation of the mitotic spindle and the alignment of chromosomes to the spindle equator [7].”
Last but extremely important aneuploidy is not a deviation from a multiple of a haploid!
Authors: Although we understand that the definition of aneuploidy is tricky and still debatable in the field, we have decided to follow the recent definition of aneuploidy from a review paper by Ben-David and Amon (Ben-David&Amon, 2020, Nature Reviews Genetics), we now cite at that specific part: “If improper kinetochore attachments are not resolved by error correction mechanisms [13], the outcome is often chromosome mis-segregation and aneuploidy, a state of chromosome number that is not a multiple of a haploid complement [14], both of which are associated with multiple congenital diseases and various types of cancers [15].”.
The sentence although…line 4 is not clear.
Authors: We have rewritten that sentence to be clearer: “In majority of higher eukaryotes chromosomes attach to spindle microtubules (MTs) by kinetochores, large macromolecular complexes that assemble specifically on the centromere of each chromosome [4].”.
Reviewer 2 Report
This is a review article for polar chromosomes. While there are many reviews for chromosome segregation, kinetochores, spindle checkpoint, there are few review articles that focus on behavior of polar chromosomes. On this viewpoint, this article is very good. In addition, the manuscript is well written and covers entire topics on polar chromosome. Before publication of this manuscript, if authors correct some minor points, this would be improved.
My comments are followings.
- There is no MCAK in Figure 3D. However, authors mentioned that MCAK is required for efficient removal of kinetochore attachments to the lateral walls of the MT (Figure 3D). It would be better to add MCAK into Figure 3D.
- Authors explained that experiments with the monopole spindle should be cautious. This is nice statement. However, explanation a bit confused me. Both CENP-E and Kid are required for chromosome congression. Then, if CENP-E was perturbed, chromosomes should be close to pole. I do not fully understand why chromosome are close to the equators upon CENP-E/Ndc80 double depletion. In addition, as CENP-E and Kid are important for similar direction, phenotype (distance between the pole and chromosome) should be similar. Why CENP-E and Kid depletion showed different phenotype. Although I might misunderstand mechanisms, more detail explanation or additional figures might help understanding.
- For Figure 6, there is CENP-T, but there is no explanation for CENP-T. For CENP-T, authors should cite following papers. For CENP-T-DNA interaction, they should cite Hori et al (Cell, 2008: PMID: 19070575) and for CENP-T-Ndc80C interaction, they should cite Nishino et al (EMBO J., PMID: 23334297). In this Figure, there is no CENP-C. This paper (Hara et al., Nature Cell Biol., 2018. PMID: 30420662) which should be cited demonstrated that the CENP-T pathway is more major than the CENP-C. If authors want to include CENP-C in this Figure, they should describe that Aurora B phosphorylation of Dsn1 (Mis12 complex) enhances binding pf CENP-C to the Mis12 complex (Petrovic et al., Cell, 2016, PMID: 27881301).
Author Response
Rebuttal letter
We thank the referees for their insightful comments. In the rebuttal letter we have marked the new text in blue.
Reviewer #2:
2) Comments and Suggestions for Authors
This is a review article for polar chromosomes. While there are many reviews for chromosome segregation, kinetochores, spindle checkpoint, there are few review articles that focus on behavior of polar chromosomes. On this viewpoint, this article is very good. In addition, the manuscript is well written and covers entire topics on polar chromosome. Before publication of this manuscript, if authors correct some minor points, this would be improved.
Authors: We thank the reviewer for the nice comments on the manuscript.
My comments are followings.
There is no MCAK in Figure 3D. However, authors mentioned that MCAK is required for efficient removal of kinetochore attachments to the lateral walls of the MT (Figure 3D). It would be better to add MCAK into Figure 3D.
Authors: We thank the reviewer for this suggestion. MCAK is now incorporated in Figure 3D that was further modified to better explain the presented model. The text part was also modified to better reflect the presented points: “During this gradual process, laterally attached kinetochores rarely detach in the presence of CENP-E, while kinesin-13 MCAK is required for efficient removal of kinetochore attachments to the lateral walls of the MT (Figure 3D) [96].”
Authors explained that experiments with the monopole spindle should be cautious. This is nice statement. However, explanation a bit confused me. Both CENP-E and Kid are required for chromosome congression. Then, if CENP-E was perturbed, chromosomes should be close to pole. I do not fully understand why chromosome are close to the equators upon CENP-E/Ndc80 double depletion. In addition, as CENP-E and Kid are important for similar direction, phenotype (distance between the pole and chromosome) should be similar. Why CENP-E and Kid depletion showed different phenotype. Although I might misunderstand mechanisms, more detail explanation or additional figures might help understanding.
Authors: We thank the reviewer for this comment. We modified the text part to better explain why CENP-E is influencing alignment differently depending on stability of spindle microtubules, primarily based on the data from Iemura&Tanaka, 2015, Nature Communications paper, that first pointed out that perplexity in CENP-E function, and data from Gudimchuk et al., 2013, Nature Cell Biology paper that showed how CENP-E could track dynamic microtubules: “These results imply that CENP-E plus end directed motor activity is dominant in conditions where stable MTs are present, such as during late prometaphase, while during early prometaphase CENP-E could suppress chromosome congression by causing kinetochores to track short and unstable MTs [53, 110]”.
For Figure 6, there is CENP-T, but there is no explanation for CENP-T. For CENP-T, authors should cite following papers. For CENP-T-DNA interaction, they should cite Hori et al (Cell, 2008: PMID: 19070575) and for CENP-T-Ndc80C interaction, they should cite Nishino et al (EMBO J., PMID: 23334297). In this Figure, there is no CENP-C. This paper (Hara et al., Nature Cell Biol., 2018. PMID: 30420662) which should be cited demonstrated that the CENP-T pathway is more major than the CENP-C. If authors want to include CENP-C in this Figure, they should describe that Aurora B phosphorylation of Dsn1 (Mis12 complex) enhances binding pf CENP-C to the Mis12 complex (Petrovic et al., Cell, 2016, PMID: 27881301).
Authors: We thank the reviewer for these detailed suggestions. We introduced new sentences explaining how Ndc80, as a part of KMN network, is connected to inner kinetochore by the DNA-interacting CENP-T, and cited the suggested papers: “The Ndc80 complex is essential for the establishment of end-on attachments of kinetochores to MTs [4,113,135]. The Knl1/Mis12 complex/Ncd80 complex (KMN network) is connected to the inner kinetochore primarily by the DNA-interacting CENP-T [136–138] (Figure 6).”. To simplify the figures, we did not introduce CENP-C in Figure 6.
Round 2
Reviewer 1 Report
the authors have clearly improved the figures and the figure legends to clarify the mechanisms proposed, have rendered the text more readable so that the paper is now much improved and can be published.
I would suggest one very minor changes to better reflect microscopic observations of mitosis. in figure 4A the chromosokinesins on chromosome arm push chromosome away in the direction of the microtubule growth, so the chromosome arms should be depicted having the direction of the microtubule, so almost parallel to them
Author Response
Rebuttal letter
We thank the referee for his comments that improved the quality of the figures.
Reviewer #1:
1) The authors have clearly improved the figures and the figure legends to clarify the mechanisms proposed, have rendered the text more readable so that the paper is now much improved and can be published.
Authors: We thank the reviewer for his positive comments.
I would suggest one very minor changes to better reflect microscopic observations of mitosis. in figure 4A the chromokinesins on chromosome arm push chromosome away in the direction of the microtubule growth, so the chromosome arms should be depicted having the direction of the microtubule, so almost parallel to them
Authors: We have adapted all the parts of Figure 4 to incorporate the point raised by the reviewer, redrawing all chromosomes that are depicted to be under polar ejection force.